# Effect of Rolling Temperature on Microstructure and Properties of Al-Mg-Li Alloy

**DOI:** 10.3390/ma15217517

**Published:** 2022-10-26

**Authors:** Weiwei Li, Mingdong Wu, Daihong Xiao, Lanping Huang, Wensheng Liu, Sai Tang

**Affiliations:** Science and Technology on High-Strength Structural Materials Laboratory, Central South University, Changsha 410083, China

**Keywords:** Al-Mg-Li alloy, rolling temperature, aerospace materials, microstructure, properties

## Abstract

In this study, the effects of hot-rolled processes at different temperatures (420 °C, 450 °C, and 480 °C) and subsequent solid solution and aging treatments on the microstructure, mechanical properties, and corrosion properties of Al-Mg-Li alloys with trace Sc and Zr addition were investigated. The aging treatment of rolled sheets after solid solution treatment could obtain Al_3_Li particles and Al_3_(Sc, Zr)/Al_3_Li core–shell particles to improve the mechanical properties of Al-Mg-Li alloy products effectively. The results showed that, as the rolling temperatures increased from 420 °C to 480 °C, the alloy’s ultimate tensile strengths and yield strengths increased, while the corrosion resistance decreased. The increase in rolling temperature increased the precipitation-free zone (PFZ) width of the alloy, which undermined the corrosion resistance of the alloy. Moreover, elevating the hot rolling temperature changes the texture strength of the alloy. Particularly in the 480 °C hot-rolled sample, the decrease in the Brass texture strength and the increase in the S texture and Copper texture strength led to an increase in the Taylor factor (*M*). The increase in rolling temperature also raised the number density of the Al_3_(Sc, Zr)/Al_3_Li core–shell particles. The presence of such particles not only inhibits grain growth but also changes the strength mechanism from dislocation cutting to Orowan bypassing. Due to the combination effect of grain morphology, texture evolution, and precipitation behavior, the 480 °C hot-rolled sample had the highest properties.

## 1. Introduction

The addition of small amounts of lithium to aluminum gives the alloy an unparalleled combination of mechanical properties, which has elevated aluminum–lithium alloys to the highest position as the preferred structural materials for aerospace applications [1,2].

Compared to Al-Li alloys with Cu, Al-Mg-Li alloys are of great interest in the aerospace industry, especially in the design of ultra-lightweight aircraft components that require weight reduction [3,4], due to their low density, high specific strength, great welding properties, and excellent corrosion resistance, such as 1420 and 1424 Al-Li alloys [5,6,7]. Currently, rolling deformation is the main method to enhance the mechanical properties of Al-Mg-Li alloys and their processing properties. Nevertheless, it shows low plasticity and poor machinability at room temperature and is prone to cracking during deformation. Therefore, improving the rolling performance by optimizing the rolling process is crucial to enhance the properties of Al-Mg Li alloys.

The rolling temperature and pressing rate are the crucial parameters affecting the performance of Al alloys [8]. Liu et al. investigated the effect of different hot rolling temperatures on the mechanical properties and grain morphology evolution of aluminum alloys [9]. They found that controlling the texture structure can effectively improve the alloy’s mechanical strength. Zuiko et al. rolled aluminum alloys at low temperatures in a liquid nitrogen environment to retard the microstructure evolution and enhance the strength of the alloy [10]. In addition, Liu et al. and Zhao et al. also studied the effect of cold and hot rolling on the texture development of aluminum-x alloys [11,12]. Previous studies have demonstrated that the texture is intentionally controlled by alloy processing techniques such as hot rolling, cold rolling, and annealing, which can substantially improve the properties of aluminum alloys [13,14]. Hence, the investigation of the effect of rolling temperature on the texture evolution in Al-Mg-Li alloys is of great importance.

Al-Mg-Li alloy is a precipitation-strengthened material, and its main precipitates are Al_3_Li (δ′), Al_2_MgLi, and AlLi. Since Al_2_MgLi and AlLi precipitates are located at grain boundaries, they exert little effect in strengthening alloys [15]. The main strengthening phase in Al-Mg-Li alloys is Al_3_Li. The investigation of the effect of rolling temperature on the behavior of the δ′ precipitated phase in Al-Mg-Li alloys is necessary. δ′, an ordered strengthening phase with a large volume fraction in Al-Mg-Li alloy, is precipitated during the aging process [16,17]. Therefore, the aging treatment of rolled sheets or formed parts after solid solution treatment could obtain the desired microstructure to enhance the mechanical properties of Al-Mg-Li alloy products effectively.

Although there are some previous studies on the rolling of Al-Mg-Li alloys, there is a lack of study on the grain morphology, texture evolution, and precipitation behavior of Al-Mg-Li alloys at different rolling temperatures. Thereby, we still know little about how the rolling temperature affects the Al-Mg-Li alloy’s properties. This investigation aims to combine the effects of grain morphology, texture evolution, and precipitation behavior on properties at different rolling temperatures. In this study, the mechanical properties were obtained by tensile testing using a mechanical testing machine, the precipitated phases were observed by transmission electron microscopy (TEM), and the grain configuration and texture of specimens at different rolling temperatures were characterized by electron backscatter diffraction (EBSD). The effects of rolling temperatures (420 °C, 450 °C, and 480 °C) on the microstructure and properties of Al-Mg-Li alloys were systematically investigated.

## 2. Experimental Procedures

### Experimental Methods

In this work, Al-Mg-Li alloy was obtained by a melting and casting method. Inductively coupled plasma emission spectrometry (ICP-AES, iCAP7600, Thermo Fisher Scientific, Waltham, MA, USA) was used to measure the chemical composition of Al-Mg-Li alloy, as shown in Table 1.

The as-cast Al-Mg-Li alloy was preheated at 450 °C for 80 min for extrusion into sheets, and the extrusion ratio of sheets was 9, as shown in Figure 1. Rectangular samples with dimensions of 100 mm × 40 mm × 10 mm were wire-cut from the middle of the Al-Mg-Li alloy sheet for subsequent hot rolling. The rolling speed was 50 mm/s. The reduction ratio was 80%. After being preheated at 420 °C, 450 °C, and 480 °C for 2 h, the thickness of sheets was reduced from 10 mm to 2 mm after being hot-rolled. The samples subjected to hot rolling at 420 °C, 450 °C, and 480 °C were marked as S420, S450, and S480, respectively. After rolling, the specimens were subjected to a solid solution treatment at 450 °C for 2 h in a resistance furnace, and then quenched in water at room temperature, and finally aged at 145 °C for 48 h in an electric thermostatic drying oven.

Tensile samples with a scale length of a 6 mm (width) × 2 mm (thickness) cross-section and a 30 mm gauge length were cut in the middle of these sheets along the rolling direction, as shown in Figure 2. The room-temperature tensile test was performed with an Instron 3369 mechanical tester at a speed of 2 mm/min. Moreover, an extensometer was attached to the tensile samples to determine strain and total elongation. Three tests were performed on each type of rolled sample, and the average value was used as the test result.

The intergranular corrosion (IGC) tests studied the corrosion behavior of the alloy. For the IGC test, the samples were polished with 1.5 μm Al_2_O_3_; then, we used ethanol to clean them, and finally used an electric breeze to dry them. Samples for the intergranular corrosion test were immersed in 30 g/Ls NaCl + 10 mL/L HCl solution at 35 ± 2 °C for 24 h. The corrosion damage on the cross-section of the sample was characterized by optical microscopy (OM, Leica DM4500P, Wizz, Germany). The dimensions of corrosion specimens are shown in Figure 3.

The microstructures of all samples were taken from the rolling direction (RD)–transverse direction (TD) plane. The samples were ground with 240-, 600-, and 2000-grit silicon carbide sandpaper and polished with 1.5 μm Al_2_O_3_ polishing solution. After mechanical polishing, the samples were electrolytically polished with 10% HClO_4_ and 90% C_2_H_5_OH at a temperature of 25 °C and a voltage of 20 V for 3 s to obtain higher-quality EBSD maps. The ZEISS EVO MA10 scanning electron microscope and an OXFORD EBSD detector (at 25 kV) were used to carry out the EBSD tests. The CHANNEL 5 software was used to analyze data. The number, density, size, and distribution of precipitates were characterized by a transmission electron microscope (TEM, JEM-2100F, JEOL, Tokyo, Japan). For TEM observations, samples were ground to a thickness of 80 μm. A twin-jet electrolytic thinning instrument was used to electropolish the specimens with a mixed solution of 25% HNO_3_ and 75% CH_3_OH at −30 °C.

## 3. Results

### 3.1. Grain Structure and Texture Characterization

Further information about the grain morphology and its spatial distribution can be obtained by EBSD. Figure 4 compares the grain morphology and size differences between the three samples. After aging treatment of the rolled samples, many of the grains are still elongated, indicating that some deformation still exists. In addition, there are many small-sized and equiaxed grains with different orientations in contrast to the elongated grains. The coupling of these is formed a sandwich structure with the large-sized grains. The increase in rolling temperature makes the average grain size of the samples increase firstly and then decrease. The average grain size of the S420 sample is 3.8 μm, which is less than that of the S450 (4.5 μm) and S480 (3.9 μm) samples. Therefore, an increase in hot rolling temperature also causes grain growth and the recrystallization of the deformed grains.

Figure 5 shows the statistical analysis of the grain boundary orientation distribution of the aged samples. As the rolling temperature increases, the proportion of low-angle grain boundaries increases and then decreases. The percentage of low-angle grain boundaries for the S420, S450, and S480 samples is 66.3%, 76.2%, and 65.4%, respectively. The ratio of low-angle grain boundaries in the S480 sample is smaller than that of S420 and S450, while the average grain boundary angle is larger than that of S420 and S450.

Figure 6 shows the distribution of samples’ recrystallized grains, substructure, and deformation organization at different rolling temperatures after aging treatment by analyzing the EBSD data. In the recrystallization diagrams, different colors represent different grain states. Blue, yellow, and red indicate recrystallized grains, substructure, and deformation, respectively. After aging treatment of S420 specimens, there is a large amount of substructure combination with a small amount of recrystallized grains and some deformation organization, indicating that the degree of recrystallization is low. After aging treatment of S450 samples, deformed grains and sub-grains are the dominant structure, and the recrystallization degree decreases steeply. As the rolling temperature increases to 480 ℃, recrystallization grains increase significantly for compensation, implying a significant increase in the recrystallization degree. In summary, as the rolling temperature increases, the sub-structure and deformation grains are significantly reduced, and the degree of recrystallization is significantly increased.

Plastic deformation causes the grains to rotate in the direction appropriate to the external force and form textures. The aluminum–lithium alloy typically forms brass {011} <211>, copper {112} <111>, and S {123} <634> textures after rolling [13,14]. The recrystallized textures will also be formed during the subsequent heat treatment. We have analyzed the EBSD data of the RD–TD planes of samples after solid solution and aging treatment [18], and the obtained orientation distribution function (ODF) plots for ϕ2 = 0°, ϕ2 = 45°, and ϕ2 = 65° are shown in Figure 7a–c. Cube {001} <100>, R-Cube {001} <110>, and Goss {011} <100> textures are easily formed during the subsequent aging process due to recrystallization [14,15]. The result also shows that the typical rolling textures (S, Copper, and Brass textures) exist in all three samples after rolling, but the strength of the textures varies. Sample S420 has intense brass and S textures, while the Goss and Cube textures are weak and negligible. Sample S480 has strong Copper and S textures. As the rolling temperature increases, the strength of the Brass texture gradually decreases, while the strength of the S and Copper textures gradually increases.

Studying the composition of alloy textures can be achieved by labeling various textures in EBSD diagrams. Typical textures (Copper, S, Brass) of three specimens with different rolling temperatures after aging treatment are marked by CHANNEL 5 software, and the content fraction of each texture is calculated within an angular tolerance of 20°. As shown in Figure 8, the volume fractions of the texture contained in the three samples are distinct, showing significant differences in the evolution of the texture for the different rolling temperature samples. Notably, the Brass texture’s volume fraction is higher in all three samples. As shown in Figure 9, the S420 samples had the highest volume percentage of the Brass and R-Cube textures and the lowest volume percentages of the S and Copper textures. The volume percentages of the Brass and R-Cube textures decreased as the rolling temperature increased. In contrast, the volume percentages of the S and Copper textures increased and then decreased as the rolling temperature increased.

### 3.2. Precipitation Behavior

Adding Sc to Al-Li alloys produces outstanding effects, such as grain refinement, substructural strengthening, and precipitation phase strengthening due to Al_3_Sc dispersion [19,20]. Moreover, the addition of Zr can bring about similar effects [21,22]. Therefore, the addition of Sc and Zr can substantially improve the mechanical properties of Al-Li alloys [23,24]. Al_3_Sc and Al_3_Zr have approximately +1.5% and +1% lattice mismatch with the matrix, respectively. In comparison, δ′ has only approximately −0.08% mismatch. Therefore, to reduce the strain energy and surface energy, the Al_3_Sc/α-Al matrix interface and the Al_3_Zr/α-Al matrix interface can act as heterogeneous nucleation sites for the δ′ phase. Accordingly, the Al_3_(Sc, Zr)/Al_3_Li core–shell structure composite phase may form after complete wetting [24,25,26]. Figure 10 shows the HAADF images and EDS plots of the Al_3_(Sc, Zr)/Al_3_Li composite particles in the S420 sample. The aggregation of Zr and Sc elements is visible in the EDS figure.

Figure 11 shows the TEM dark-field images of the precipitates in the grain interior and at the grain boundary, and the inserted figure shows selected electron diffraction speckle patterns of the S420, S450, and S480 samples taken on the [110]_Al_ axis. It is observed in Figure 11a,c,e that spherical δ′ phase and Al_3_(Sc, Zr)/Al_3_Li core–shell structure composite phases coexisted in all samples. However, the hot rolling temperature can change the size and number density of these particles. For the S420 sample after aging treatment, it contains the fine and uniformly distributed δ′ phase with an average particle dimension of approximately 12.1 nm. On the contrary, a small amount of Al_3_(Sc, Zr)/Al_3_Li with a large size of 34.2 nm is presented. As the rolling temperature increases to 450 ℃, the number density of the Al_3_(Sc, Zr)/Al_3_Li core–shell structure composite phase is higher than that of the S420 sample. The inhibitory effect of Al_3_(Sc, Zr) on the growth of the δ′ phase makes the average particle dimension of the δ′ phase in the S450 sample smaller than that of the S420 sample. The average particle dimensions of the δ′ phase and the composite particles are around 8.9 nm and 22.1 nm in the S450 sample, respectively. The Al_3_(Sc, Zr)/Al_3_Li core–shell structure composite phases have the highest number density when the rolling temperature reaches 480 ℃. Therefore, the number density of Al_3_(Sc, Zr)/ Al_3_Li core–shell structure composite phases can be increased significantly with the increasing rolling temperature.

Figure 11b–d show the related TEM dark-field images of the grain boundary morphology after aging treatment for the S420, S450, and S480 samples. With the increase in rolling temperature, the PFZ width near the grain boundary increases firstly and then decreases. The highest PFZ width is 41.1 nm for the S450 sample, and the lowest is 22.3 nm for the S420 sample. The corrosion performance of the specimens is greatly influenced by the width of the PFZ. The wider the PFZ, the worse the corrosion performance of the sample. The results of the subsequent intergranular corrosion test of each sample are basically consistent with the influence of the PFZ width on the corrosion performance of the samples.

### 3.3. Tensile Properties

Figure 12 shows the samples’ stress–strain curves and histograms of the tensile properties at different rolling temperatures after aging treatment. With the increasing rolling temperature, the alloys’ yield strength (YS) and ultimate tensile strength (UTS) increase gradually. The UTS values of the S420, S450, and S480 samples were 477 MPa, 486 MPa, and 496 MPa, respectively. In contrast to UTS, the YS of the specimens increases significantly as the rolling temperature increases. The YS of S480 is higher than that of S420 (35 MPa) and S450 (37 MPa), while the elongation shows the opposite trend. The percentage of low-angle grain boundaries for the S420, S450, and S480 samples is 66.3%, 76.2%, and 65.4%, respectively (Figure 5). The S450 sample has the largest number of low-angle grain boundaries, and the higher the proportion of low-angle grain boundaries, the better the plasticity of the sample, so the S450 sample has the best elongation.

### 3.4. Intergranular Corrosion Properties

Figure 13 and Figure 14 show the samples’ corrosion morphology and related maximum corrosion depths at different rolling temperatures. The S420 sample and S480 sample present the features of local corrosion. The maximum corrosion depths of the S420 sample, S450 sample, and S480 sample are 27.3 μm, 36.1 μm, and 28.5 μm, respectively. There is little difference between the S420 and S480 samples’ intergranular corrosion depths. The S450 sample’s intergranular corrosion is the most serious. The corrosion edge produces a large piece of shedding. The corrosion mode shows the characteristics of pitting. It is significantly different from the S420 sample and the S480 sample. As the rolling temperature rises, the intergranular corrosion resistance gradually becomes worse.

## 4. Discussion

### 4.1. The Effect of Rolling Temperature on the Mechanical Properties of the Alloy

Incomplete dynamic recrystallization occurred during the aluminum alloys’ hot rolling process. Meanwhile, the hot rolling temperature greatly impacted the formation and growth of recrystallized grains. The mechanical properties of alloys are affected by different rolling temperatures. The YS of alloys is affected due to the well-known strengthening mechanisms, including grain refinement, work hardening, solid solution strengthening, and precipitation strengthening [15]. As the hot rolling temperature increases, the average grain size of alloys becomes larger. The difference in average grain dimension causes a change in strength through grain refinement strengthening. The equation of the Hall–Petch relationship describes the grain boundary strengthening [27,28,29]:(1)$$\sigma_{GB}=k_{y}d^{-0.5}$$
where $k_{y}$ is a constant with a value of approximately 0.08 Mpa·m^0.5^ [27,28,29], and d is the average grain size. The average grain sizes of the three samples in Figure 4 are 3.8 μm, 4.5 μm, and 3.9 μm, respectively. Based on Equation (1), the strength derived from grain boundary $\sigma_{GB}$ can be calculated as 41.1 MPa, 37.7 MPa, and 40.5 MPa. Therefore, the grain boundary strengthening impact of the S420 sample was the strongest among the three specimens, followed by the S480 sample, and it was lowest for the S450 sample.

The orientation and textures of the tensile axis can determine the Taylor factor (*M*). The Taylor factor can be determined by several methods based on the number of active slip systems, such as the Sachs and Taylor models, etc. Table 2 shows the values of *M* for many textures based on different models in the rolling direction [30]. Three models are used to calculate the average *M* values for different rolling temperature samples in the rolling direction, and the results are presented in Table 3. As the rolling temperature increases, the *M* value in the rolling direction of the sample increases slightly. The superposition of different strengthening effects determines the YS of polycrystalline materials, which can be calculated by Equation (2) [30,31]:(2)$$\Delta \sigma =M\Delta \tau +\sigma_{GB}$$
where $\sigma_{GB}$ is the strength factor determined by grain boundary strengthening, and ∆*τ* is the critical shear stress of the grain. ∆*τ* can be calculated as [30,31]
(3)$$\Delta \tau =\tau_{0}+\tau_{ss}+\sqrt{\tau_{D}^{2}+\tau_{P}^{2}}$$
where $\tau_{0}$ represents the internal stress of pure aluminum, and $\tau_{ss}$ represents the contribution of the solid solution strength to stress. In contrast, the heat-treated aluminum alloys can neglect the solid solution strength. $\tau_{D}$ represents the contribution of dislocations to stress, and $\tau_{P}$ represents the contribution of precipitation strengthening to stress. The increase in rolling temperature causes a notable increase in the number density of Al_3_(Sc, Zr)/Al_3_Li core–shell structure composite phases. The composite particle causes the dislocation cutting mechanism to change from a shearing mechanism to an Orowan bypassing mechanism. The values of $\tau_{D}$ and $\tau_{P}$ are elevated, and the combined factors of the three increase ∆τ, thus enhancing the strength.

As the rolling temperature increases, the evolution of the texture structure occurs, and the value of Taylor factor M increases, which increases the strength. The difference in the grain boundary strengthening effect of different rolling temperature samples is insignificant, which causes the strength difference to be within 4 MPa. The change in the precipitation phase causes the dislocation cutting mechanism to change and the precipitation strengthening effect to increase, so the evolution of the texture structure and the shift in precipitation phase play a significant role in the strength increase.

### 4.2. The Effect of Rolling Temperature on the Corrosion Performance of the Alloy

The intergranular corrosion test (Figure 13) shows that as the rolling temperature increases, the corrosion resistance of the alloy decreases. As mentioned above, the rolling temperature affects the alloy’s microstructure evolution and the related properties in the subsequent solid solution aging process. In parallel, the corrosion behaviors of the alloy can be influenced by the precipitate in the grains and at the grain boundaries (Figure 11) [32]. Numerous studies show that the corrosion properties of aluminum alloys are dominated by grain boundary precipitates, which depends on the grain boundary structure [32,33]. According to the literature [34], the corrosion resistance of aluminum alloys can be enhanced by increasing the low-angle grain boundaries to inhibit the alloy’s precipitation of GBPs. Moreover, the width of the PFZ also has a significant impact on the corrosion performance of alloys. The wider the PFZ, the worse the corrosion resistance.

This is mainly due to its electrochemical nature, i.e., the corrosion occurs due to the potential difference between the intermetallic phase and the aluminum matrix caused by chemical inhomogeneity.

The higher the rolling temperature, the higher the degree of recrystallization and the larger the recrystallized grain size. Due to the high interface energy of recrystallized grains, more large-angle grain boundaries will be formed between the grains. The aging precipitation phase will be easily nucleated preferentially, forming continuously distributed coarse particles and causing PFZ broadening. The grain boundary energy of the subcrystalline organization is lower than that of recrystallized organization, and the enrichment of the aging precipitation phase on the subcrystalline boundary is lower than that of the large angle-grain boundary. It is not easy to form a continuous grain boundary precipitation phase, and the width of the grain boundary PFZ is narrower. The unrecrystallized grains exhibit better resistance to intergranular corrosion compared to recrystallized grains. The dislocation density also affects the corrosion resistance of the alloy. Dislocation is a crystal defect; the greater the density of dislocation, the worse the corrosion resistance of the alloy. Moreover, with the increase in rolling temperature, the number density of Al_3_(Sc, Zr)/Al_3_Li composite particles in the alloy increases. The composite particle enhanced the located effect on dislocations and increased the dislocation density during deformation. In summary, as the rolling temperature rises, the intergranular corrosion resistance gradually becomes worse.

## 5. Conclusions

The potential effects of the hot rolling temperature on Al-Mg-Li alloys’ tensile properties, microstructure, and corrosion behavior were investigated. The main conclusions are as follows:(1)With the increase in rolling temperature, the ultimate tensile strengths of the S420, S450, and S480 samples are 477 MPa, 486 MPa, and 496 MPa, respectively. The yield strengths are 377 MPa, 375 MPa, and 412 MPa, respectively, and the elongations are 5%, 10%, and 6%. Thus, the strength of the alloy gradually increases with the increase in the hot rolling temperature, while there is a partial loss of elongation.(2)With the increase in rolling temperature, the maximum grain boundary corrosion depths of the S420, S450, and S480 samples are 27.3 μm, 36.1 μm, and 28.5 μm. The corrosion resistance of the samples is in the order of S420 > S480 > S450. With the increase in rolling temperature, the degree of recrystallization and the recrystallized grain size of the sample become higher. More large-angle grain boundaries are formed between recrystallized grains. The large-angle grain boundary is the corrosion-prone part. It causes the PFZ width of the alloys to increase. Meanwhile, the precipitated phase affects the dislocation density. The dislocation density of the sample increases with the increasing rolling temperature, which worsens the corrosion resistance of alloys. As the rolling temperature rises, the intergranular corrosion resistance gradually becomes worse.(3)As the rolling temperature increases, the strength of the Brass texture in the alloy decreases, while the strength of S and Copper textures increases. The evolution of the texture increases the value of Taylor factor M; thus, the strength of the alloy increases. The increasing rolling temperature also raises the number density of the sample’s Al_3_(Sc, Zr)/Al_3_Li core–shell structure composite phase. The presence of this composite particle not only inhibits grain growth but also changes the dislocation cutting mechanism to the Orowan bypassing mechanism, and the combined effect of the two increases the strength of the material.

## Figures and Tables

**Figure 1 materials-15-07517-f001:**
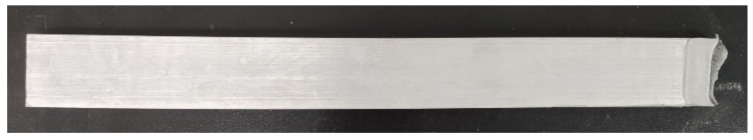
Al-Mg-Li alloy extrusion sheet.

**Figure 2 materials-15-07517-f002:**
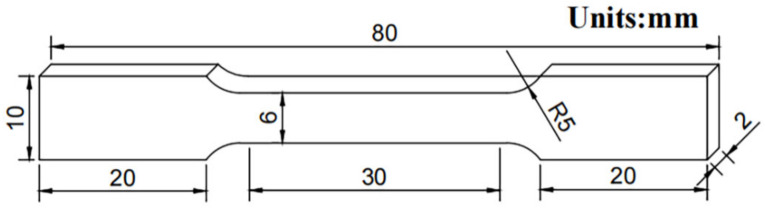
The dimensions of tensile specimens.

**Figure 3 materials-15-07517-f003:**
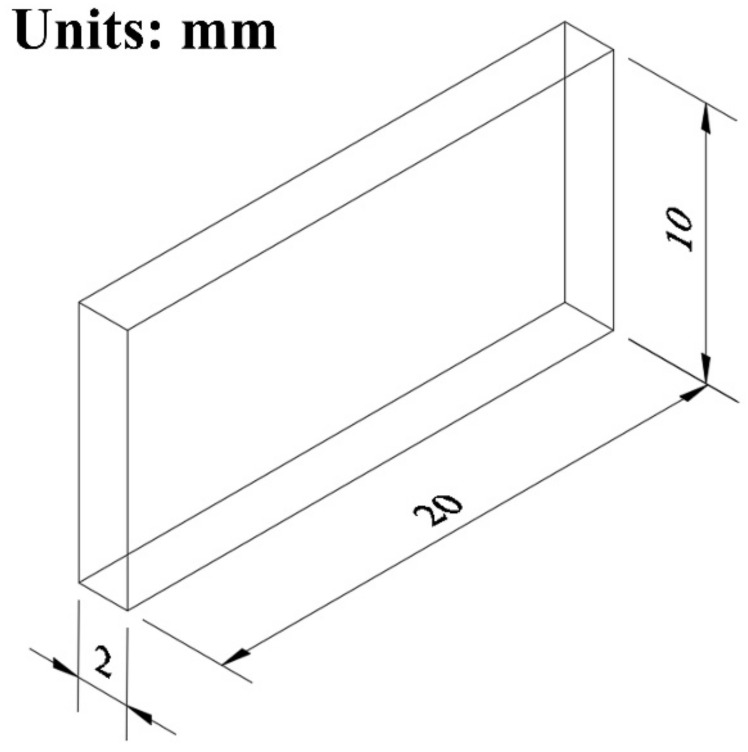
The dimensions of corrosion specimens.

**Figure 4 materials-15-07517-f004:**
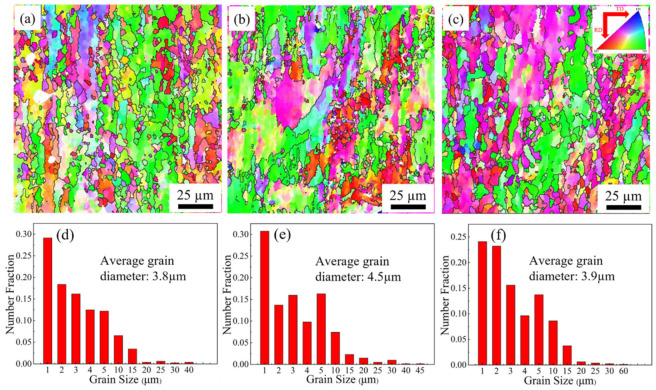
IPF diagrams and corresponding grain size distributions of samples at different rolling temperatures after aging treatment: S420 (**a**,**d**), S450 (**b**,**e**), S480 (**c**,**f**).

**Figure 5 materials-15-07517-f005:**
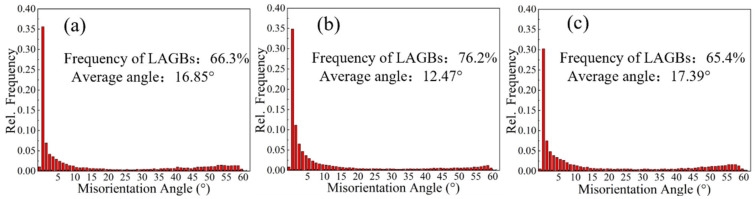
Statistical plots of grain boundary orientation differences of samples at different rolling temperatures after aging treatment: S420 (**a**), S450 (**b**), S480 (**c**).

**Figure 6 materials-15-07517-f006:**
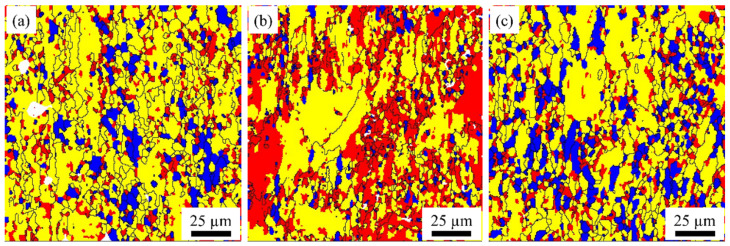
Recrystallization diagrams of samples at different rolling temperatures after aging treatment: S420 (**a**), S450 (**b**), S480 (**c**).

**Figure 7 materials-15-07517-f007:**
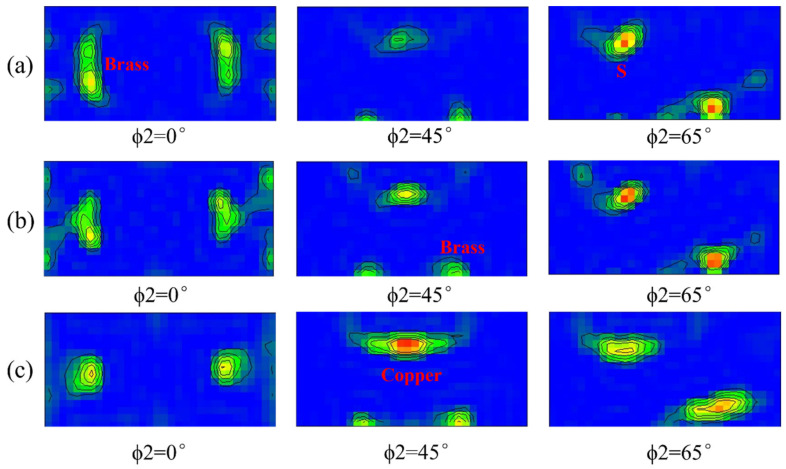
ODF diagrams of samples at different rolling temperatures after aging treatment: S420 (**a**), S450 (**b**), S480 (**c**).

**Figure 8 materials-15-07517-f008:**
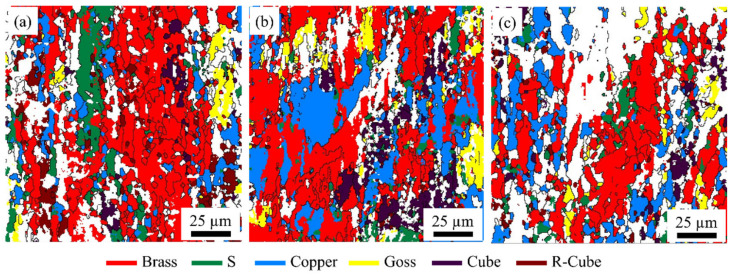
Distribution of crystal orientation of samples at different rolling temperatures after aging treatment: S420 (**a**), S450 (**b**), S480 (**c**).

**Figure 9 materials-15-07517-f009:**
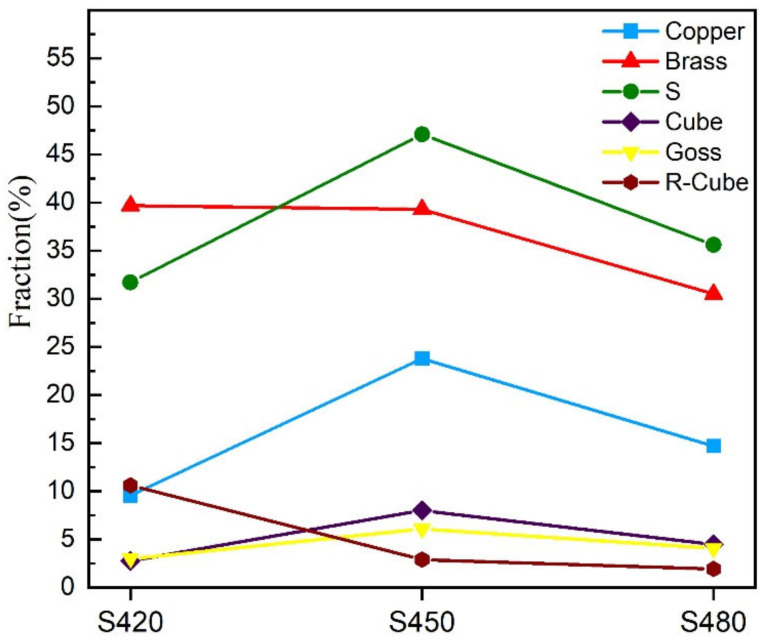
Graphs of the main texture changes of samples at different rolling temperatures after aging treatment.

**Figure 10 materials-15-07517-f010:**
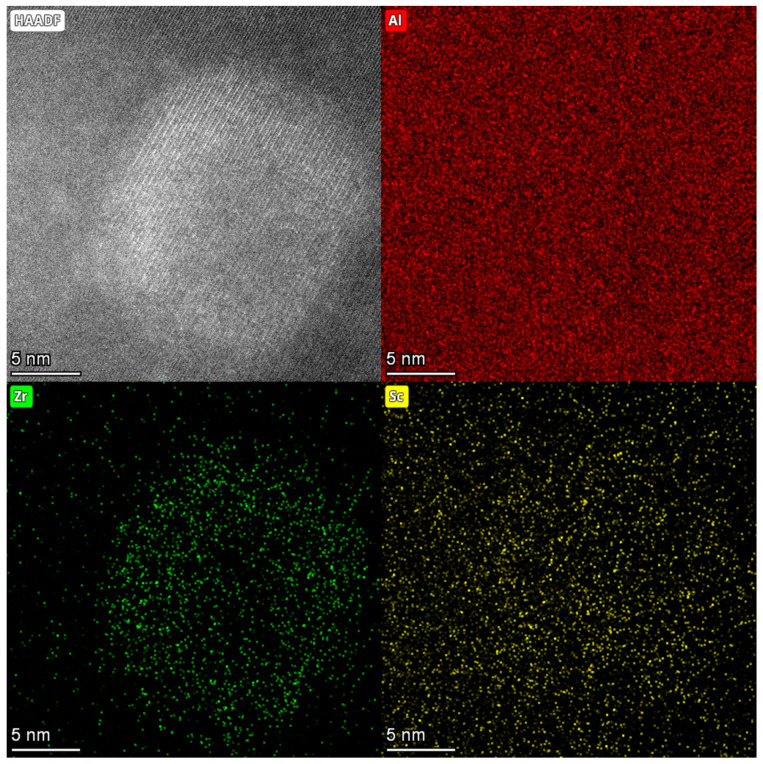
HAADF images and EDS plots of the Al_3_(Sc, Zr)/Al_3_Li core–shell particles in the S420 sample.

**Figure 11 materials-15-07517-f011:**
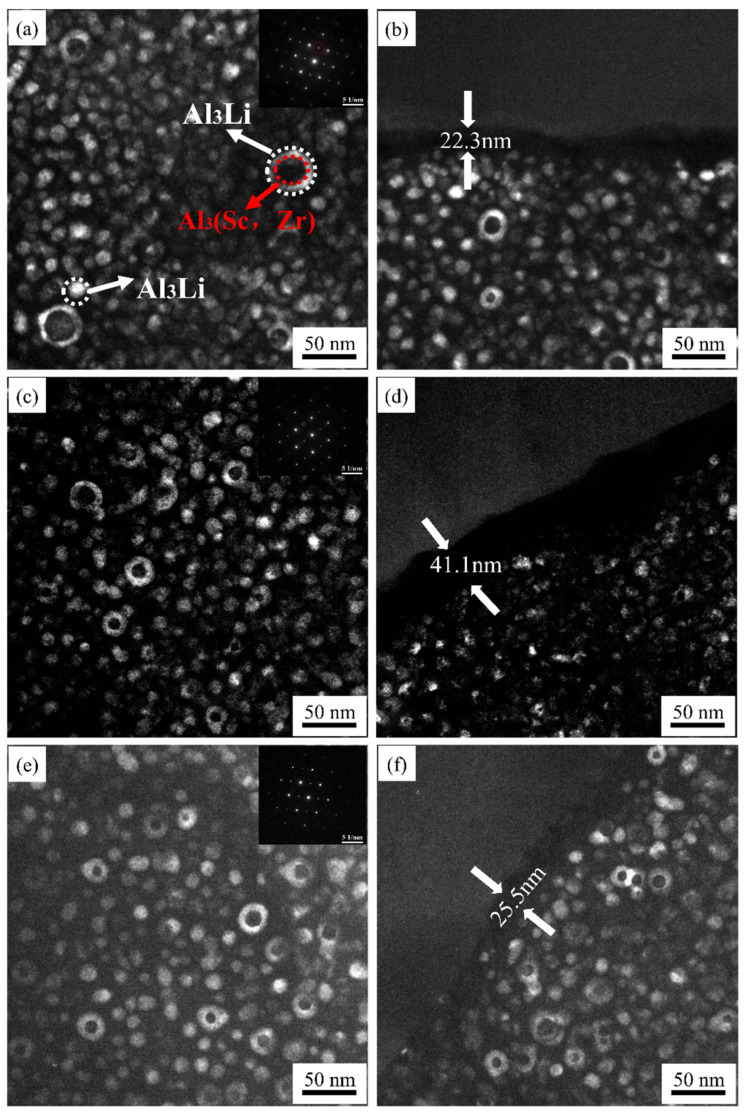
TEM dark-field images and electron diffraction spot maps of samples at different rolling temperatures after aging treatment: S420 (**a**,**b**), S450 (**c**,**d**), S480 (**e**,**f**).

**Figure 12 materials-15-07517-f012:**
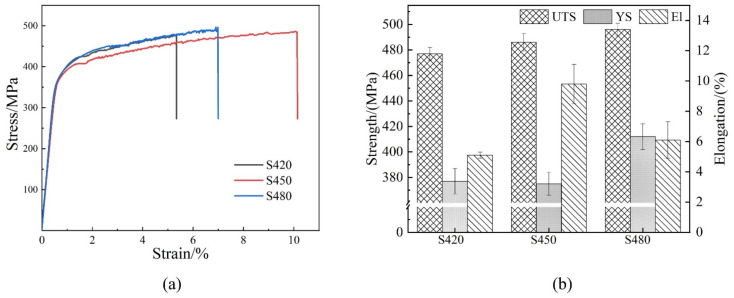
Room-temperature mechanical properties of samples at different rolling temperatures after aging treatment: stress–strain curve (**a**), comparison of ultimate tensile strength, yield strength, and elongation (**b**).

**Figure 13 materials-15-07517-f013:**
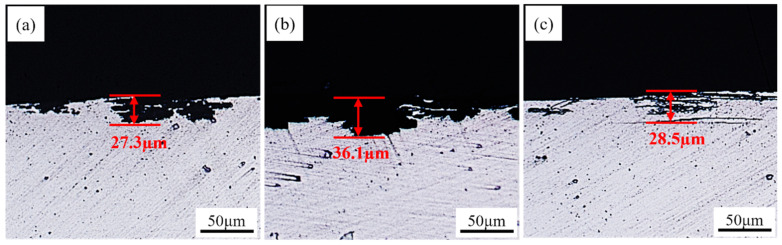
Plots of maximum intergranular corrosion depth of samples at different rolling temperatures after aging treatment: S420 (**a**), S450 (**b**), S480 (**c**).

**Figure 14 materials-15-07517-f014:**
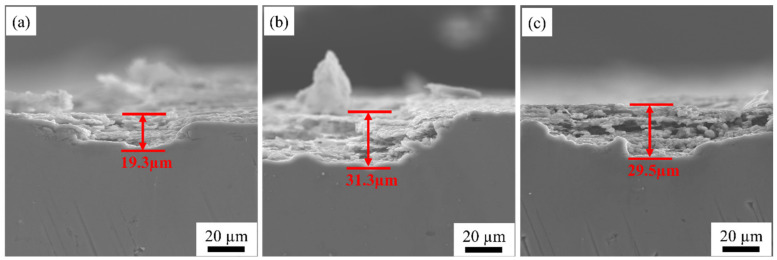
SEM image of corrosion morphology of samples at different rolling temperatures after aging treatment: S420 (**a**), S450 (**b**), S480 (**c**).

**Table 1 materials-15-07517-t001:** Exact chemical composition of Al-Mg-Li alloy (wt. %).

Mg	Li	Zn	Sc	Zr	Al
4.75	1.77	0.53	0.06	0.11	Bal

**Table 2 materials-15-07517-t002:** M values for macroscopic loads in the rolling direction for different rolling and recrystallization textures [30].

	5 Systems (Taylor Model)	1 System (Sachs Model)	3.5 Systems (According to Hutchinson’s Model)
Typical rolling textures			
Copper	3.70	3.04	3.44
S	3.33	2.50	2.97
Brass	3.17	2.44	2.94
Typical recrystallization textures			
Goss	2.45	2.45	2.45
Cube	2.45	2.45	2.45
Texture-free FCC classical models	3.07	2.24	2.60

**Table 3 materials-15-07517-t003:** Average M values for macroscopic loading in the rolling direction of three different samples.

Different Samples	5 Systems (Taylor Model)	1 System (Sachs Model)	3.5 Systems (According to Hutchinson’s Model)
S420	3.23	2.51	2.96
S450	3.25	2.57	2.99
S480	3.26	2.57	2.99

## Data Availability

The raw/processed data required to reproduce these findings cannot be shared at this time as the data also form part of an ongoing study.

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
