# Peer review of "Effect of Rolling Temperature on Microstructure and Properties of Al-Mg-Li Alloy"

_materials, 2022, doi:10.3390/ma15217517_

Round 1
Reviewer 1 Report
The paper is well written and interesting for researchers working in the field of Al-Mg-Li alloy.
In the Methodology section, it would be good to write in more detail at what speed the rolling was
carried out, with what degree of deformation, and how the cast alloy was extruded.
The study was carried out at a fairly high level, the conclusions are justified and the article needs minimal revision.
Author Response
Thank you for your comments concerning our manuscript. Please see the attachment.

Reviewer 2 Report
1. The elongation values of the samples (obtained with different rolling temperature) where the UTS values almost did not change, varied considerably. What is the mechanism behind this should be discussed in the tensile properties section.
2. Intergranular corrosion tests have been carried out, but only corrosion surface cross-sectional images are given. Corrosion tendency at metal crystallite boundaries is not shown. The mechanism of intergranular corrosion needs to be clearly explained. Corrosion effect on the boundaries' SEM images etc. should be supported.
3. Numerical values should be given in the conclusions section and the results emphasizing the contribution to the literature should be included.
4. In the line 303; it is not Fig 9, it is Fig 10.
5. The effect of the dislocation density, grain boundary type and the grain has a big impact on IGC. What can the authors comment on this subject according to the results they have obtained for the materials fabricated?
Author Response

(The authors gave the same response as above.)

Reviewer 3 Report
Dear Authors,
well done with your efforts.
Kindly note these issues:
1-Abstract
Passive voice should be used instead of active voice. We should be omitted.
Introduction
2- Line 48, So would be replaced with (Hence,.....)
Results
3-which instrument is used for Figure 3? it is not mentioned in the text. SEM, TEM,..?
4- line 218, As we know, should be replaced with a passive voice.
Reference
5- references' styles should be reconsidered. Just for example, reference no: 10, 11, 12, ..
Author Response

(The authors gave the same response as above.)

Reviewer 4 Report
This article has presented the effect of rolling temperature on the mechanical, microstructure, and corrosion behavior of Al-Mg-Li alloy. However, certain shortcomings must be addressed before further processing.
[1]. Correct the first sentence of the abstract and write the abbreviation first and then use it further like PFZ in the abstract.
[2]. What is meant by aluminum-x alloy at line 45?
[3]. There is a conflict in the statements in lines 33-34 and 50-52. Lithium-based alloys are considered sensitive to the heat-treatment for improving mechanical behavior, while lines 33-34 highlight that rolling deformation is the main method for non-heat-treatable alloys.
[4]. The same issue of abbreviation at lines 97-98.
[5]. Include the pictures of the working sheet of Al-Mg-Li alloy along with specimens of tensile and corrosion testing.
[6]. The authors have performed the aging treatment as described on page 4 in Results Section. The effect of aging must be included in the abstract, and further information and execution must be in the Section of Experimental Procedures along with the details of temperature and time cycles.
[7]. Are these standard textures, Copper, S, Brass, and R-Cube textures, for the analysis of grain texture? Can you please link with some ASTM standards or any other standard or reference?
[8]. What is the HAADF image in the Figure 7 caption?
[9]. It is written in lines 228-229 that the tensile strength increase gradually but it’s hardly observed from Fig. 9b.
[10]. The authors have done some work and fail to highlight the exact novelty in the last paragraph of the introduction.
Author Response
Thank you for your comments concerning our manuscript.Please see the attachment.

Round 2
Reviewer 4 Report
The following Points need to be addressed in the revised manuscript:
For Point 7, include some relevant references in the manuscript.
For 9, reduce the scale-bar difference of the y-axis for a clear-cut increment in UTS.
Author Response
Thank you for comments concerning our manuscript. Please see the attachment.
